# Investigation of NPB Analogs That Target Phosphorylation of BAD-Ser99 in Human Mammary Carcinoma Cells

**DOI:** 10.3390/ijms222011002

**Published:** 2021-10-12

**Authors:** Swamy Savvemala Girimanchanaika, Dukanya Dukanya, Ananda Swamynayaka, Divya Maldepalli Govindachar, Mahendra Madegowda, Ganga Periyasamy, Kanchugarakoppal Subbegowda Rangappa, Vijay Pandey, Peter E. Lobie, Basappa Basappa

**Affiliations:** 1Laboratory Chemical Biology, Department of Studies in Organic Chemistry, University of Mysore, Manasagangotri, Mysore 570006, India; swamynayak010@gmail.com (S.S.G.); dukanya4@gmail.com (D.D.); 2Department of Studies in Physics, University of Mysore, Manasagangotri, Mysore 570006, India; anandas@physics.uni-mysore.ac.in (A.S.); mahen_xrdlab@yahoo.com (M.M.); 3Department of Chemistry, Bangalore University, Bangalore 560056, India; divyamg03@gmail.com (D.M.G.); ganga.periyasamy@gmail.com (G.P.); 4Tsinghua Berkeley Shenzhen Institute, Tsinghua Shenzhen International Graduate School, Tsinghua University, Shenzhen 518055, China; vijay.pandey@sz.tsinghua.edu.cn; 5Institute of Biopharmaceutical and Health Engineering, Tsinghua Shenzhen International Graduate School, Tsinghua University, Shenzhen 518055, China; 6Institution of Excellence, University of Mysore, Manasagangotri, Mysore 570006, India; rangappaks@gmail.com; 7Shenzen Bay Laboratory, Shenzhen 518055, China

**Keywords:** BAD phosphorylation, Petasis reaction, lead optimization, drug design, human mammary carcinoma

## Abstract

The design and development of a small molecule named NPB [3-{(4(2,3-dichlorophenyl)piperazin-1-yl}{2-hydroxyphenyl)methyl}-N-cyclopentylbenzamide], which specifically inhibited the phosphorylation of BAD at Ser99 in human carcinoma cells has been previously reported. Herein, the synthesis, characterization, and effect on cancer cell viability of NPB analogs, and the single-crystal X-ray crystallographic studies of an example compound (4r), which was grown via slow-solvent evaporation technique is reported. Screening for loss of viability in mammary carcinoma cells revealed that compounds such as 2[(4(2,3-dichlorophenyl)piperazin-1-yl][naphthalen-1-yl]methyl)phenol (4e), 5[(4(2,3-dichlorophenyl)piperazin-1-yl][2-hydroxyphenyl)methyl)uran-2-carbaldehyde (4f), 3[(2-hydroxyphenyl][4(p-tolyl)piperazin-1-yl)methyl)benzaldehyde (4i), and NPB inhibited the viability of MCF-7 cells with IC_50_ values of 5.90, 3.11, 7.68, and 6.5 µM, respectively. The loss of cell viability was enhanced by the NPB analogs synthesized by adding newer rings such as naphthalene and furan-2-carbaldehyde in place of N-cyclopentyl-benzamide of NPB. Furthermore, these compounds decreased Ser99 phosphorylation of hBAD. Additional in silico density functional theory calculations suggested possibilities for other analogs of NPB that may be more suitable for further development.

## 1. Introduction

BCL-2-associated death promoter (BAD) is a modulator of apoptosis, which when unphosphorylated directly interacts with BCL-w, BCL-2, and BCL-xL, amongst other proteins [1]. Phosphorylation of BAD (pBAD) is required for its heterodimerization with 14-3-3 protein and promotion of cancer cell survival [2,3]. Specifically, the phosphorylation of human (h)BAD at Ser-75, Ser-99, and Ser-118 is required to promote cancer cell survival [4]. Human BAD is phosphorylated independently at Ser-75 and Ser-99 by RAS/RAF/MAPK and PI3K/AKT/mTOR pathways, respectively [5]. In addition, all three PIM kinase family members may also phosphorylate hBAD on multiple sites but require prior phosphorylation of Ser-75 or Ser-99 and function as rescue kinases for BAD phosphorylation upon inhibition of RAS/RAF/MAPK and PI3/AKT/mTOR pathways [5,6,7,8]. Hence, as a common downstream cell survival mediator of both the RAS/RAF/MAPK and PI3K/AKT/mTPR pathways, phosphorylated BAD has been demonstrated to be critically involved in cancer development, progression, and therapeutic resistance. [5]. Therefore, pharmacological inhibition of BAD phosphorylation may be of utility to enhance therapeutic outcomes in oncology. Towards this goal, a novel bioactive small molecule called NPB [N-cyclopentyl-3{(4(2,3-dichlorophenyl)piperazin-1-yl}{2-hydroxyphenyl} methyl) benzamide] was previously identified, which specifically inhibited the phosphorylation of hBAD on Ser-99 in various carcinoma cells independent of kinase activities [9,10]. Furthermore, NPB enhanced the efficacy of cisplatin in ovarian carcinoma and synergized with AZD5363, an AKT inhibitor in cisplatin resistant ovarian cancer [3]. Herein, the synthesis, characterization, and efficacy of newer NPB analogs with replacement of different substituents (R_1_, R_2_, and R_3_) is reported (Figure 1) [11,12,13,14,15,16].

## 2. Results and Discussion

The NPB analogs were synthesized based on the Petasis borono−Mannich reaction using N-substituted-piperazines, salicylaldehyde, and various boronic acids as the nucleophilic reagent [17,18,19,20,21]. In this multicomponent reaction, the iminium ion formation occurs initially, which reacts with boronic acid to form a tetracoordinate boronate in situ intermediate, and eventually the product formation occurs by intramolecular delivery of the organic group to iminium carbon (Figure 1). The structures of all NPB analogs were characterized by LCMS, ^1^H NMR, and ^13^C NMR spectroscopic techniques (Table 1, refer supplementary spectra of all the compounds).

Furthermore, the crystal structure of an example compound was determined using a slow evaporation technique [22]. Compound 4r emerged as a crystal and therefore its X-ray intensity data were recorded at a temperature of 293K on a Bruker AXS kappa Apex2 CCD Diffractometer, with a fine-focus sealed tube radiation source (MoKα) and 0.71073 Å wavelength. The procedure and reduction of the data set was accomplished using SAINT PLUS. SHELXS and SHELXL programs were adopted to solve and refine the structure, respectively [23,24]. The geometrical calculations, molecular figures, and crystal packing were generated and visualized by PLATON and MERCURY software, respectively (Table 2) [25]. CCDC number 2027110 contains full crystallographic data of **4r** and is available online at the Cambridge crystallographic data center. The ORTEP diagram was obtained for compound **4r** (Figure 2a). The crystal packing of the structure (Figure 2b) revealed the importance of the oxygen atom in the hydroxy group (i.e., present in all **4a-s**) and the chlorine atom in the para position of the phenyl ring (i.e., especially in the more potent compounds **4e** and **4f**), which confirms their participation in potential hydrogen bond formation via **O** and **Cl** atoms to form a three-dimensional supramolecular hydrogen-bonded network. This reflects the importance for compound stability and its role in interaction with other molecules. Such information is helpful in structure and activity relationships that provide a solid basis for structure-based optimization in the future design of further compounds. Structural analysis revealed that both phenyl and thiophene rings exhibited planar conformation, while the six-membered piperazine is in a chair conformation exhibiting puckering parameters: amplitude (Q) = 0.5535 Å, Theta = 4.61°, and Phi = 154.0425°. The planarity conformation of the phenyl and thiophene rings allows for the partial overlapping of aromatic rings, which play an important role in biological activity [26]. In addition, the hybridization of the C-C bond which is considered as one of the most important and common chemical elements, especially for organic connections, is usually formed by s and p orbitals of the second shell in carbon and lead to the formation of different bonds. Among several types of this hybridization, the 4r molecule exhibited **sp^2^** hybridization that formed with two single bonds and one double bond between three atoms showing a 120° angle value between bonds. This type of hybridization was observed in phenyl and thiophene rings. On the other side, the piperazine ring exhibited **sp^3^** hybridization in which the carbon atom is bonded to four other atoms forming only a single bond. Here, 1s orbital and 3p orbitals in the same shell of an atom combine to form four new equivalent orbitals. The presence of different types of hybridization enhances the bond strength, stability, and reactivity of the molecule.

Since NPB was previously reported to inhibit the viability of various carcinoma cells, its analogs were tested for their ability to inhibit mammary carcinoma cell viability using the reported protocol [27,28]. The results of the study revealed that the compounds such as 2[(4(2,3-dichlorophenyl)piperazin-1-yl][naphthalen-1-yl]methylphenol (**4e**), 5[(4(2,3-dichlorophenyl)piperazin-1-yl][2-hydroxyphenyl]methylfuran-2-carbaldehyde (**4f**), 3[(2-hydroxyphenyl)4(p-tolyl)piperazin-1-yl]methylbenzaldehyde (**4i**) as well as **NPB** inhibited the viability of MCF-7 cells with an IC_50_ values of 5.9, 3.11, 7.68, and 6.53 µM, respectively (Table 1). Interestingly, the compounds **4e, 4f, 4i,** and NPB inhibited the viability of normal breast cell MCF10A, with higher IC_50_ values of 33.8, 61.4, 28.5, and 110.6 µM, respectively. The NPB analogs synthesized by adding newer substituents such as naphthalene and furan-2-carbaldehyde in place of N-cyclopentylbenzamide of NPB slightly enhanced the loss of cell viability in MCF-7 cells. It was important to note that the dichlorophenyl group in NPB and its analogs seems quite important; however, some of the tested NPB analogs, which were synthesized by replacing the dichlorophenyl, and N-cyclopentylbenzamide group of NPB with 4-p-tolyl-group and benzaldehyde substitution, also exhibited enhanced loss of cell viability. Among 4-chloro-phenyl group containing piperazine compounds, **4b** and **4c** showed better inhibitory effects on viability of MCF-7 cells with IC_50_ values of 20.91 and 23.83 µM, whereas for the tolyl group added piperazine compounds such as **4l** and **4m,** IC_50_ values were observed to be 63.66, and 45.73 µM, respectively.

As the tested NPB analogs produced loss of cell viability to variable extents in mammary carcinoma cells, in silico density functional theory calculations were performed in order to understand the structure activity relationship of NPB analogs against the loss of cell viability (Appendix A). The highest occupied molecular orbital (HOMO) and lowest unoccupied molecular orbital (LUMO) energy values showed the electron donating and accepting ability of NPB analogs, respectively. Computed HOMO and LUMO eigen functions indicated that the donor–acceptor nature of NPB analogs localized with MOs at different regions (Figure 3 and Figure 4). Therefore, the substitutions at both ends were observed to significantly alter the electronic density levels. The computed data indicated that IC_50_ decreases with decreasing E_HOMO_ for NPB analogs, which bear electronically similar functional groups. Calculated HOMO and LUMO values are in the range of −5 to −6.5 eV and −1.5 to 2.5 eV (Table 3). Molecular electrostatic potential shows the charge separation between the two ends within the molecule. Improved loss of cell viability is observed with NPB analogs which possess similar functional groups. Hence, the NPB analogs, which possess electron donor function as three separate classes, were analyzed. In the first set, amongst the tested compounds, i.e., **4a**, **4b**, **4c** and **4r**, the compound **4b** has comparatively lower E_HOMO_ of -6.16 eV and E_LUMO_ of −2.70 eV values. Due to the relatively smaller HOMO-LUMO gaps, electronegativity and electrophilicity values, and chemical hardness of **4b**, the lower the observed IC_50_ in mammary carcinoma cells. In the second class of NPB analogs, namely 4d, 4e, 4f, 4g, and 4h, the activities are arranged in ascending order of E_HOMO_ and E_LUMO_ values, i.e., 4g < 4d < 4h < 4e < 4f, which is in accordance with their respective IC_50_ values (Figure 5). In the last class of tolyl group containing molecules such as 4i, 4j, 4k, 4l, 4m, 4n, 4o, 4p, and 4q, the molecule **4i** found to be highly effective against MCF-7 cells, which possess lower electronic factors compared to the other similar functional group tagged molecules. DFT studies predicted the activity of the molecule in line with experimental predictions. Among the molecules studied here, **4f** exhibits higher activity than other molecules.

As phosphorylation of hBAD at Ser-99 promotes cancer cell survival and NPB was reported to specifically inhibit BAD-Ser99 phosphorylation, western blot analysis was performed to evaluate the efficacy of the most active NPB analogs (**4f**, **4e**, and **4i**) on BAD-Ser99 phosphorylation in MCF-7 cells. All the tested compounds decreased the pBAD at Ser-99 without change in total BAD expression (Figure 6).

Since the active compounds (**4e**, **4f**, **4i**, and **NPB** as a comparison) displayed efficacy against MCF-7 cells, the in silico ADMET properties (8 parameters) of these compounds were determined by using vNN-ADMET online platform [29]. The results are tabulated in Table 4. The in silico analyses of active compounds predicted that the compounds (**4e**, **4f**, **4i**) would not exhibit hepatotoxicity.

## 3. Materials and Methods

Materials and reagents were purchased from commercial suppliers and used as instructed. Melting points were determined through an open capillary method using Sigma melting point apparatus (Sigma, Bangalore, India) and are uncorrected. IR spectra were recorded on Shimadzu IR spectrophotometer (Shimadzu USA manufacturing Inc., Canby, OR, USA). ^1^H NMR and ^13^C NMR spectra were recorded on Bruker/Agilent NMR spectrometer operating at 400 and 100 MHz, respectively, using TMS as internal standard; chemical shifts are in *d*. Mass spectroscopic analysis was performed on Shimadzu LC-MS. Analytical TLCs were implemented on pre-coated Merck 0.25 mm silica gel 60F254 plates using 40% ethylacetate in *n*-hexane as eluent and the spots were detected under UV light. All other chemicals were of analytical grade and were purchased from Sisco Research Laboratories (SRL, Mumbai, India).

**General procedure for synthesis of NPB analogs.** The piperazines (1eq) and salicylaldehydes (1eq) were taken in a round bottom flask and stirred with dioxane as a solvent for 10 min. After 10 min the aryl boronic acid (1eq) was added to the mixture and refluxed for 8 h on a hot plate at 90 °C with continuous stirring. After 8 h, ethyl acetate and water were added to the reaction mixture, separating the ethyl acetate layer using a separate funnel, and drying over anhydrous sodium sulphate. Ethyl acetate was evaporated to produce the product. The desired phenolic compound product was obtained by column chromatography [10].

**Characterization of 3**(**4**(**4-chlorophenyl**)**piperazin-1-yl**) (**2hydroxyphenyl**)**methyl**)**-N-cyclopen tylbenzamide** (**4a**) Off-white solid; mp 128–130 °C; 91% yield; ^1^H NMR (CDCl_3_, 400 MHz) δ:11.49 (s, 1H), 7.84 (s, 1H), 7.64–7.59 (m, 2H), 7.36 (t, *J* = 8Hz,1H), 7.19 (d, *J* = 4Hz, 2H), 7.14 (t, *J* = 8Hz, 2H), 6.96 (d, *J* = 4Hz, 1H), 6.87 (d, *J* = 8Hz, 1H), 6.79 (d, *J* = 8Hz, 1H), 6.75–6.72 (m, 1H), 6.10 (s, 1H), 4.53 (s, 1H), 4.41–4.36 (m, 1H), 3.20 (s, 4H), 2.83–2.59 (m, 4H), 2.08 (s, 2H), 1.73–1.65 (m,2H), 1.52–1.42 (m,2H), 1.34–1.26 (m, 2H); ^13^C NMR (CDCl_3_, 100 MHz) δ:166.8, 156.2, 149.4, 140.3, 135.6, 131.1, 129.3, 129.1, 128.9, 127.6, 126.3, 125.2, 124.6, 119.8, 117.8, 117.2, 114.5, 76.1, 51.9, 51.6, 49.3, 33.2. 23.8; HRMS (ESI-TOF) m/z: [M+H]^+^ calcd for C_29_H_32_ClN_3_O_2_, 490.2261; found, 490.2259.

**Characterization of 2**(**4**(**4-chlorophenyl**)**piperazin-1-yl**) (**6-methylpyridin-3-yl**)**methyl**)**phenol** (**4b**) Off-white solid; mp 210–212 °C; 88% yield; ^1^H NMR (CDCl_3_, 400 MHz) δ: 11.50 (br-s, 1H) 8.55 (s, 1H) 7.80–7.78 (m, 1H) 7.26–7.15 (m, 4H) 6.99 (d, *J* = 8Hz, 1H) 6.93 (d, *J* = 8Hz, 1H) 6.86–6.79 (m, 3H) 4.55 (s, 1H) 3.33–3.22 (m, 4H) 2.82–2.66 (m, 4H) 2.56 (s, 3H); ^13^C NMR (CDCl_3_, 100 MHz) δ:158.7, 156.2, 149.4, 149.2, 136.2, 132.0, 129.2, 129.1,125.3, 124.3,123.8, 119.8, 117.7,117.7,117.3, 73.2,51.5, 49.3, 24.1; HRMS (ESI-TOF) m/z: [M+H]^+^ calcd for C_23_H_24_ClN_3_O, 394.1686; found, 394.1682.

**Characterization of 3(4(4-chlorophenyl)piperazin-1-yl) (4(diethylamino)-2-hydroxyphenyl) methyl)-N-cyclopentylbenzamide (4c)** Brown solid; mp 178–180 °C; 81% yield; ^1^H NMR (CDCl_3_, 400 MHz) δ: 11.23 (br-s, 1H), 7.78 (s, 1H), 7.97 (d, *J* = 8Hz, 2H), 7.38–7.34 (m, 1H), 7.18 (d, *J* = 8Hz, 2H), 6.80–6.73 (m, 2H), 6.18 (s,1H), 6.08–6.01 (m, 2H), 4.44 (s, 1H), 4.39 (s, 1H), 3.30–3.27 (m, 4H), 3.25–3.19 (m, 4H), 2.62–2.52 (m, 2H), 2.06–2.10 (m, 3H), 1.71–1.48 (m, 8H), 1.12 (t, *J* = 8Hz, 6H); ^13^C NMR (CDCl_3_, 100 MHz) δ: 166.9, 157.1, 149.5, 148.9, 141.0, 135.4, 131.2, 129.9, 129.7, 129.1, 127.3, 125.9, 124.9, 117.3, 111.6, 103.5, 99.7, 75.4, 51.8, 51.4, 49.3, 44.2, 33.2, 23.8,12.7; HRMS (ESI-TOF) m/z: [M+H]^+^ calcd for C_33_H_41_ClN_4_O_2_, 561.2996; found, 561.2998.

**Characterization of 1**(**5**((**4**(**2,3-dichlorophenyl**)**piperazin-1-yl**) (**2-hydroxyphenyl**)**methyl**)**thiophen-2-yl**)**ethanone** (**4d**) White solid; mp 130–132 °C; 89% yield; ^1^H NMR (CDCl_3_, 400 MHz) δ: 11.14 (br-s, 1H) 7.60 (d, *J* = 4Hz, 1H) 7.25–7.17 (m, 4H) 7.06–6.93 (m, 3H) 6.84–6.80 (m, 1H) 4.87 (s, 1H) 3.19–3.16 (m, 4H) 2.89–2.79 (m, 4H) 2.55 (s, 3H); ^13^C NMR (CDCl_3_, 400 MHz) δ:190.5, 155.9, 150.6, 150.5, 144.4, 134.2, 132.4, 129.5, 128.9, 127.7, 127.6 (2C), 125.1, 124.0,119.8,118.7, 117.4, 70.6, 51.5, 51.2, 26.7; LCMS m/z: [M+H]^+^ calcd for C_23_H_22_Cl_2_N_2_O_2_S, 461.0857; found, 460.9520.

**Characterization of 2-**((**4-**(**2,3-dichlorophenyl**)**piperazin-1-yl**)(**naphthalen-1-yl**)**methyl**)**phenol** (**4e**)**.** Brown solid; mp 78–80 °C; 78% yield; ^1^H NMR (CDCl_3_, 400 MHz) δ:8.08 (d, *J* = 8Hz, 1H) 7.91–7.74 (m, 3H) 7.93–7.33 (m, 2H) 7.22–7.14 (m, 2H) 6.98 (d, *J* = 8Hz, 1H) 6.90–6.83 (m, 1H) 6.78–6.55 (m, 3H) 6.60–6.59 (m, 1H) 5.37 (s, 1H) 4.23 (s, 1H) 3.37–3.28 (m, 4H) 3.04–2.33 (m, 4H); ^13^C NMR (CDCl_3_, 100 MHz) δ:155.8, 154.8, 136.7, 135.1, 134.1, 131.1, 130.9, 129.3, 129.2, 128.9, 128.6, 128.4, 126.1, 125.8, 125.4, 124.1, 123.7, 120.6, 119.9, 117.0, 116.9, 74.0, 52.7, 52.5; HRMS (ESI-TOF) m/z: [M+H]^+^ calcd for C_27_H_24_Cl_2_N_2_O, 463.1343; found, 463.1340.

**Characterization of 5**(**4**(**2,3-dichlorophenyl**)**piperazin-1-yl**) (**2-hydroxyphenyl**)**methyl**)**furan-2-carbaldehyde** (**4f**) Brown solid; mp 128–130 °C; 83% yield; ^1^H NMR (CDCl_3_, 400 MHz) δ:10.06 (s, 1H) 8.20 (s,1H) 8.07–7.90 (m, 1H) 7.74–7.44 (m, 2H) 7.20–7.12 (m, 1H) 6.96–6.87 (m, 1H) 6.35–6.33 (m, 1H) 6.21–6.12 (m, 1H) 5.29 (s,1H) 4.41 (s,1H), 3.41–3.04 (m, 4H) 2.34–2.10 (m, 4H); ^13^C NMR (CDCl_3_, 100 MHz) δ: 177.6, 158.4, 156.7, 152.4,150.4, 134.1, 129.8, 128.9, 127.6, 127.5, 125.1, 121.9, 121.3, 119.8,111.6, 117.2, 112.0, 67.7, 51.2, 50.6; HRMS (ESI-TOF) m/z: [M+H]^+^ calcd for C_22_H_20_Cl_2_N_2_O_3_, 431.0929; found, 431.0924

**Characterization of 2**(**4(2,3-dichlorophenyl**)**piperazin-1-yl**) (**6-methylpyridin-3-yl**)**methyl**)**phe nol, 4g** Yellow solid; mp 208–210 °C; 83% yield; ^1^H NMR (CDCl_3_,, 400 MHz) δ:11.50 (br-s, 1H), 8.55 (s, 1H), 7.80–7.77 (m, 1H), 7.25–7.16 (m, 4H), 7.00–6.78 (m, 4H), 4.55 (s, 1H), 3.26 (s, 4H), 2.81–2.66 (m, 4H), 2.58 (s, 3H); ^13^C NMR (CDCl_3_,, 100 MHz) δ:158.7, 156.2, 149.4, 149.2, 136.1, 132.0, 129.5, 129.1,129.0,125.2, 124.3, 124.1, 124.0, 123.8, 119.8, 117.5, 117.3, 73.2,51.5, 49.3,24.1; HRMS (ESI-TOF) m/z: [M+H]^+^ calcd for C_23_H_23_Cl_2_N_3_O, 428.3542; found, 428.0576.

**Characterization of N-cyclopentyl-3(4(2,3-dichlorophenyl)piperazin-1-yl) (4(diethylamin o)-2-hydr oxyphenyl) methyl) benzamide, 4h.** White solid; mp 160–162 °C; 81% yield; ^1^H NMR (CDCl_3_, 400 MHz) δ:7.78 (s,1H) 7.58–7.56 (m,1H) 7.38–7.34 (m, 1H) 7.18 (d *J* = 8Hz, 2H) 6.80–6.73 (m, 2H) 6.18 (s, 1H), 6.09–6.00 (m, 2H) 4.44 (s, 1H) 4.41–4.34 (m, 1H) 3.40–3.19 (m, 6H) 3.19–3.10 (m, 2H); ^13^C NMR (CDCl_3_, 100 MHz) δ: 168.4, 154.6, 149.4, 137.5, 133.8, 132.5,128.9, 128.4, 128.2, 127.4, 127.0, 126.6, 125.8, 125.6, 119.2, 118.5, 110.2, 103.4, 101.2, 68.2, 51.5, 48.8, 44.8, 33.2, 29.6, 24.3, 13.1; HRMS (ESI-TOF) m/z: [M+H]^+^ calcd for C_33_H_40_Cl_2_N_4_O_2_, 595.2606; found, 595.2601.

**Characterization of 3**(**2-hydroxyphenyl**) (**4**(**p-tolyl**)**piperazin-1-yl**)**methyl**)**benzaldehyde, 4i** Brown; mp 120–123 °C; 88% yield; ^1^H NMR (CDCl_3_, 400MHz) δ: 11.66 (s, 1H), 10.03 (s, 1H), 7.97 (s, 1H), 7.83 (d, *J* = 8 Hz, 2H), 7.55–7.52 (m, 1H), 7.20–7.17 (m, 1H), 7.10 (d, *J* = 8 Hz, 2H), 7.00 (d, *J* = 8 Hz, 1H), 6.92 (d, *J* = 8 Hz, 1H), 6.86 (d, *J* = 8 Hz, 2H), 6.80–6.77 (m, 1H), 4.62 (s, 1H), 3.25 (s, 4H), 2.66 (s, 4H), 2.29 (s, 3H); ^13^C NMR (CDCl_3_, 100MHz) δ: 192.3, 156.4, 148.8, 141.2, 137.2, 134.6, 130.1, 130.0, 129.9, 129.5, 129.3, 124.7, 120.0, 117.6, 116.8, 76.2, 52.0, 50.0, 50.1, 20.7; LCMS m/z: [M+H]^+^ calcd for C_25_H_26_N_2_O_2_, 386.1994; found, 387.2418.

**Characterization of N-cyclopentyl-3**(**2-hydroxyphenyl**) (**4**(**p-tolyl**)**piperazin-1-yl**)**methyl**)**benzamide** (**4j**); Brown; mp: 114–117 °C; 87% yield; ^1^H NMR (CDCl_3_, 400 MHz) δ: 11.72 (s, 1H), 7.87 (s, 1H), 7.64 (d, 2H, *J*=8Hz), 7.39 (t, 1H, *J* = 6Hz), 7.16 (t, 1H, *J* = 6Hz), 7.10 (d, 2H, *J* = 8Hz), 6.86 (m, 5H), 6.16 (s, 1H), 4.57 (s, 1H), 4.41 (m, 1H), 3.23 (s, 4H), 2.63 (s, 4H), 2.30 (s, 3H), 2.12 (s, 2H), 1.72 (m, 4H), 1.53 (m, 2H); ^13^C NMR (CDCl_3_, 100MHz) δ: 166.8, 156.2, 148.6, 140.4, 135.5, 131.1, 129.8, 129.7, 129.3, 128.8, 127.9, 126.3, 124.7 (2C), 119.7, 117.2, 116.6, 76.1, 51.8, 49.8, 33.2, 29.7, 29.4, 23.8, 20.5; HRMS (ESI-TOF); m/z: [M+H]^+^ calcd for C_30_H_35_N_3_O_2_, 469.2729; found, 469.2726.

**2**(**pyrimidin-5-yl)** (**4**(**p-tolyl**)**piperazin-1-yl**)**methyl**)**phenol** (**4k**)**;** Off-white solid; mp: 157–161 °C; 84% yield; ^1^H NMR (CDCl_3_, 400MHz) δ:9.16 (s, 1H), 8.84 (s, 2H), 7.19 (t, 1H, *J* = 4Hz), 7.07 (d, 2H, *J* = 4Hz), 6.92 (m, 2H), 6.81 (m, 3H), 4.50 (s, 1H), 3.89 (s, 1H), 3.20 (m, 4H), 2.74 (m, 4H), 2.27 (s, 3H); ^13^C NMR (CDCl_3_, 100MHz) δ: 158.7, 157.1, 156.2, 148.5, 130.6, 130.3, 129.9, 129.9, 129.8, 128.9, 127.7, 123.2, 120.3, 117.8, 117.2, 116.8, 71.8, 51.9, 49.9, 29.8, 20.6, 14.2; HRMS (ESI-TOF) m/z: [M+H]^+^ calcd for C_22_H_24_N_4_O;360.1950 found, 360.1953.

**Characterization of 2**(**2-fluoro-3-methylpyridin-4-yl**) (**4**(**p-tolyl**)**piperazin-1-yl**)**methyl**)**phenol** (**4l**); Yellow solid; mp: 102–104 °C; 80% yield; ^1^H NMR (CDCl_3_, 400MHz) δ:8.00 (d, 1H, *J* = 4Hz), 7.54 (d, 1H, *J* = 4Hz), 7.19 (t, 1H, *J* = 4Hz), 7.10 (d, 2H, *J* = 8Hz), 6.96 (d, 1H, *J* = 8Hz), 6.92 (d, 1H, *J* = 8Hz), 6.84 (d, 2H, *J* = 4Hz), 6.78 (t, 1H, *J* = 4Hz), 4.94 (s, 1H), 3.23 (s, 6H), 2.47 (s, 4H), 2.29 (s, 4H); ^13^C NMR (CDCl_3_, 100MHz) δ: 163.4, 161.5, 156.1, 148.4, 145.0, 144.9, 129.7, 129.3, 128.6, 123.4, 120.1, 119.8, 117.5, 116.6, 68.9, 49.8, 49.6, 20.3, 11.35; HRMS (ESI-TOF) m/z: [M+H]^+^ calcd for C_24_H_26_FN_3_O, 391.2060; found, 391.2057.

**Characterization of 2**(**5,6-dimethylpyridin-3-yl**) (**4**(**p-tolyl**)**piperazin-1-yl**)**methyl**)**phenol** (**4m**)**;** Off-white solid; mp: 147–159 °C; 86% yield; ^1^H NMR (CDCl_3_, 400MHz) δ:11.66 (s, 1H), 8.34 (s, 1H), 7.56 (s, 1H), 7.15 (t, 1H, *J* = 4Hz), 7.07 (d, 2H, *J* = 8Hz), 6.94 (d, 1H, *J* = 4Hz), 6.89 (d, 1H, *J* = 8Hz), 6.81 (d, 2H, *J* = 4Hz), 6.75 (t, 1H, *J* = 4Hz), 4.48 (s, 1H), 3.21 (s, 4H), 2.62 (s, 3H), 2.26 (d, 6H, *J* = 8Hz); ^13^C NMR (CDCl_3_, 100MHz) δ: 157.4, 156.2, 148.6, 146.4, 132.6, 129.7, 129.1, 128.8, 124.4, 119.6, 117.2, 116.6, 73.2, 49.8, 29.6, 22.2, 20.4, 19.3; HRMS (ESI-TOF) m/z: [M+H]^+^ calcd for C_25_H_29_N_3_O, 387.2311; found, 387.2316.

**Characterization of 2**(**1H-pyrazol-4-yl**) (**4**(**p-tolyl**)**piperazin-1-yl**)**methyl**)**phenol** (**4n**)**;** Off-white solid; mp: 151–153 °C; 83% yield; ^1^H NMR (CDCl_3_, 400MHz) δ: 7.59 (s, 2H), 7.15 (t, 1H, *J* = 8Hz), 7.05 (d, 2H, *J* = 8Hz), 6.94 (d, 1H, *J* = 8Hz), 6.87 (d, 1H, *J* = 8Hz), 6.80 (d, 2H, *J* = 8Hz), 6.74 (t, 1H, *J* = 4Hz), 4.63 (s, 1H), 3.14 (d, 4H, *J* = 8Hz), 2.67 (s, 4H), 2.25 (s, 3H); ^13^C NMR (CDCl_3_, 100MHz) δ: 156.5, 148.6, 133.7, 129.8, 129.6, 128.8, 128.6, 125.4, 119.3, 118.0, 116.8, 116.6, 65.6, 50.7, 49.8, 29.6, 20.4; HRMS (ESI-TOF) m/z: [M+H]^+^ calcd for C_21_H_24_N_4_, 332.2001; found, 332.2005.

**Characterization of 2**(**2-aminopyrimidin-5-yl**) (**4**(**p-tolyl**)**piperazin-1-yl**)**methyl**)**phenol** (**4o**); Yellow solid; mp: 131–134 °C; 91% yield; ^1^H NMR (CDCl_3_, 400MHz) δ:8.32 (s, 2H), 7.16 (t, 1H, *J* = 6Hz), 7.05 (d, 2H, *J* = 8Hz), 6.88 (t, 2H, *J* = 6Hz), 6.77 (m, 3H), 5.44 (s, 2H), 4.36 (s, 1H), 3.19 (s, 4H), 2.65 (s, 4H), 2.25 (s, 3H); ^13^C NMR (CDCl_3_, 100MHz) δ: 162.8, 158.6, 156.3, 148.4, 129.9, 129.7, 129.0, 128.7, 123.8, 122.2, 119.7, 117.3, 116.6, 51.3, 49.8, 20.4; HRMS (ESI-TOF) m/z: [M+H]^+^ calcd for C_22_H_25_N_5_O, 375.2059; found, 375.2056.

**Characterization of 5**(**2-hydroxyphenyl**) (**4**(**p-tolyl**)**piperazin-1-yl**)**methyl**)**furan-2-carbaldehyde** (**4p**); Off-white solid; mp 94–97 °C; 87% yield; ^1^H NMR (CDCl_3_, 400MHz) δ: 10.9 (s, 1H), 9.62 (s, 1H), 7.22 (t, 1H, *J* = 4Hz), 7.07 (d, 2H, *J* = 4Hz), 6.95 (d, 1H, *J* = 8Hz), 6.91 (d, 1H, *J* = 8Hz), 6.81 (t, 1H, *J* = 4Hz), 6.62 (s, 1H), 4.82 (s, 1H), 3.20 (s, 4H), 2.75 (s, 4H), 2.27 (s, 3H); ^13^C NMR (CDCl_3_, 100MHz) δ: 177.9, 158.8, 157.1, 152.7, 148.8, 130.5, 130.1, 130.1, 129.2, 121.7, 120.1, 117.9, 117.0, 112.3, 68.0, 51.3, 50.2, 20.8; HRMS (ESI-TOF) m/z: [M+H]^+^ calcd for C_23_H_24_N_2_O_3_, 376.1787; found, 376.1783.

**Characterization of 2**(**6-methylpyridin-3-yl**) (**4**(**p-tolyl**)**piperazin-1-yl**)**methyl**)**phenol** (**4q**): Yellow solid; mp: 152–154 °C; 93% yield; ^1^H NMR (CDCl_3_, 400MHz) δ:11.61 (s, 1H), 8.51 (s, 1H), 7.75 (d, 1H, *J* = 8Hz), 7.17 (d, 1H, *J* = 8Hz), 7.12 (d, 1H, *J* = Hz), 7.07 (d, 2H, *J* = 8Hz), 6.94 (d, 1H, *J* = 12Hz), 6.89 (d, 1H, *J* = 8Hz), 6.81 (d, 2H, *J* = 12Hz), 6.76 (t, 1H, *J* = 8Hz), 4.51 (s, 1H), 3.21 (s, 4H), 2.62 (s, 4H), 2.53 (s, 3H), 2.27 (s, 3H); ^13^C NMR (CDCl_3_, 100MHz) δ: 158.7, 156.4, 149.3, 148.7, 136.2, 132.2, 129.9, 129.8, 129.2, 129.1, 124.5, 123.9, 119.8, 117.3, 116.7, 73.3, 51.7, 49.9, 24.2, 20.5; HRMS (ESI-TOF) m/z: [M+H]^+^ calcd for C_24_H_27_N_3_O, 373.2154; found, 373.2151

**Characterization of 1**(**5**(**4**(**4-chlorophenyl**)**piperazin-1-yl**) (**2-hydroxyphenyl**)**methyl**)**thiophen-2-yl**)**ethanone** (**4r**): Off-white solid; ^1^H NMR (CDCl_3_, 400 MHz) δ: 10.98 (br-s,1H) 7.60 (d, *J* = 4Hz,1H) 7.27–7.20 (m, 4H) 7.04–7.02 (m, 1H) 6.94 (d, *J* = 8Hz, 1H) 6.87–6.81 (m, 3H) 4.83 (s, 1H) 3.22–3.33 (m, 4H) 2.83–2.55 (m, 4H) 2.55 (s, 3H); ^13^C NMR (CDCl_3_, 100 MHz) δ:190.4, 155.9, 150.5, 149.3, 144.3,132.4, 129.6, 129.1, 128.9, 127.6, 125.3, 123.9, 119.9, 117.5, 117.4, 70.5, 51.1, 49.3, 26.7; LCMS m/z: [M+H]^+^ calcd for C_23_H_23_ClN_2_O_2_S, 427.1247; found, 427.0009.

**Alamar Blue assay:** The potency of title compounds against MCF-7 cells was determined using the Alamar Blue assay, following the procedure described earlier [30,31,32,33,34,35,36,37,38]. Compounds were dissolved in DMSO at 10 mg/mL concentration and stored at −20 °C. The dilutions were diluted in culture medium before treatment. A total of 10 × 10^3^ cells/well were plated in 96-well plates. After 6 h of plating, the cells were treated with different concentrations of compound in triplicates. Reagent (20 µL of 5 mg/mL) was added to the cells during the last 4h of their time point for 48 h as per manufacturer’s instructions. The medium was removed from the wells 4 h after the reagent addition, after which the absorbance was measured at 590 nm in an enzyme-linked immunosorbent assay reader.

**In silico DFT calculations:** The DFT calculations for the molecules (**4a-4r**) were carried out using B3LYP hybrid functional with 6-31 + g (d) all electron basis set utilizing the Gaussian 09 package [39]. All structures were optimized without any restraints. Partial Mulliken charges were calculated using the same level of theory to determine the charge distribution in the system. The electronic properties such as chemical hardness [η = (LUMO–HOMO)/2], electronegativity [χ =−(HOMO+LUMO)/2], chemical potential [µ = (H + L)/2], and electrophilicity index (ω = μ^2^/2η) were calculated using the energies of highest occupied molecular orbital (HOMO) and lowest unoccupied molecular orbital (LUMO). Chemical hardness (η) was used as a tool to understand the chemical reactivity of the molecular system. The concept of electronegativity (χ) is introduced as the power of an atom in a molecule to attract electrons onto itself. Electrophilicity (ω) is proposed as a measure of lowering of energy due to maximal electron flow between donor and acceptor. Conclusively, the HOMO-LUMO (Δ) gap establishes the correlation between chemical structure and biological activity.

**Western blot analysis:** Western blot analysis was carried out by using the earlier reported protocol. pBad (Ser136: equivalent to human BADSer99) and hBAD antibodies were procured from Cell Signaling and similarly mouse anti-β-ACTIN from Santa Cruz Biotechnology [9].

## 4. Conclusions

In summary, from the synthesis of a series of novel NPB analogs using the Petasis reaction, based on their efficacy against mammary carcinoma cells, compounds such as 2(4(2,3-dichlorophenyl)piperazin-1-yl) (naphthalen-1-yl)methyl)phenol (**4e**)**,** 5(4(2,3-dichlorophenyl)piperazin-1-yl) (2-hydroxyphenyl)methyl)furan-2-carbaldehyde (**4f**)**,** and 3(2-hydroxyphenyl) (4(p-tolyl)piperazin-1-yl)methyl)benzaldehyde (**4i**) were found to inhibit the viability of mammary carcinoma cells. In addition, the crystal structure of the compound **4r,** which was grown via a slow-solvent evaporation technique, is reported. Furthermore, we identified that compounds **4f, 4e** and **4i** decreased the phosphorylation of hBAD-Ser99. Such studies using NPB analogs will contribute to the lead optimization process of BAD phosphorylation inhibitors in oncology.

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
