# Peer review of "Investigation of NPB Analogs That Target Phosphorylation of BAD-Ser99 in Human Mammary Carcinoma Cells"

_ijms, 2021, doi:10.3390/ijms222011002_

Round 1
Reviewer 1 Report
In this paper, the authors synthesized and described 19 new analogues of NPB, a compound previously described as a promising new anticancer derivative.
In my mind, the aim of the study isn’t clearly mentioned except in the title “optimization” (improve NPB’s activity, its physicochemical properties, better understand its mechanism of action … ?).
The SAR discussion should be more developed to show the importance of each substituent (R1, R2 and R3) and their probably additional effect. The position of the Cl of the phenyl for R3 should be developed, its position 2, 3 and/or 4 seeming to be important for the activity. It could highlight compounds 4e and 4f which seem to be slightly better than NPB (whose IC50 on MCF7 cells isn’t reminded). Moreover, compare 4-Cl and 4-CH3 would be interesting too (4p, 4f and this one), to be sure that 4-Cl is worse. And so, for 4i, the 2,3-dichloro could be better…
In the DFT studies part, no reference is cited as well as no explanation of the principle neither the goal is developed. What about the conclusion of these observations? Could you then predict better structures?
The part on the hybridization could be shorter, and the conformation of the piperazine more developed.
What about the toxicity of these compounds? And their Lipinski score?
Why did you choose 4s for X ray and crystal packing, the only non-active analogue (and with a 4-Cl)? Be careful, note than 4r in table 1 is the same than 4d (4s doesn’t exist in the experimental section, 4r is mentioned in the conclusion…).
The introduction is well documented, but some sentences need to be improved.
For a better reading, please mention the structure of NPB, and add its previously reported activity on MCF, as it is the starting point of this new SAR (we can then compare it immediately). For a better comprehension, could you please pick table 1 and store the same colors as in the scheme 1, with the substituents in the same position than compound 4 of scheme 1 (be careful, there is an inversion between R1 and R3 in the legend of scheme 1). Same comment for figures 2 and 3.
Please, propose a new sentence for l 46, to prevent the repetition of phosphorylation and analogues.
L148 check the order, with 4r (the right one?)
L149 it’s a tolyl, not a benzyl…
L163 evaluate and confirm? please conclude
A careful re-reading is necessary (ex l54 “cticially”, l 63 “. .”, l67 “, ,”…some words are cut, numbers not in bold). The experimental part is not enough rigorously described: some 1H are missing (20 H for cpd 4h, 4 H for cpds 4i and 4m, 5 H for cpd 4m without forgetting the form (name cut, spaces, punctuations, bold, only some compounds mention “characterization of” (the name is enough).
Moreover, for the 1H description, I’m surprised by the coupling value J =4Hz. I’m OK for the thiophene, but it in my mind, it can’t be representative of two vicinal protons in a phenyl, if not, could you please justify. In most of your compounds, I don’t really see the para system, usually d, 2H, J~8.0Hz / d, 2H, J~8.0Hz due to the symmetry (note the number after the comma, especially when using a 400 MHz spectrometer to help in the assignment). Could you recheck or justify, please, as a for the t integrating for 2H (I don’t think they are equivalent)? And due to the complexity of the piperazine, I don’t think that its protons are singlets…
In my opinion, 13C NMR spectra are recorded on a 100 MHz, and all the 13C must be mentioned. Anyway you can precise 2 if 2 carbons have the same chemical shift. Moreover, the abbreviations for the NMR signals’ multiplicity are lacking in the description of material and methods.
You should cite the patent : Small molecules, including 2-[(phenyl)(piperazin-1-yl)methyl]phenol derivatives and analogs, as inhibitors of Bcl-2-associated death promoter (bad) phosphorylation
By: Lobie, Peter Edward; Pandey, Vijay Kumar; Rangappa, Kanchugarakoppal Subbegowda; Bassappa, Aalundi; Mohan, Chakrabhavi Dhananjaya; Rangappa, Shobith; Srinivasa, Venkatachalaiah World Intellectual Property Organization, WO2018194520 A1 2018-10-25
Author Response
Reply to the Reviewers Comments
Reviewer 1 comments
- In this paper, the authors synthesized and described 19 new analogues of NPB, a compound previously described as a promising new anticancer derivative. In my mind, the aim of the study isn’t clearly mentioned except in the title optimization” improve NPB’s activity, its physicochemical properties, better understand its mechanism of action … ?).
Response: We have revised the manuscript considering the above suggestion.
- The SAR discussion should be more developed to show the importance of each substituent (R1, R2 and R3) and their probably additional effect. The position of the Cl of the phenyl for R3should be developed, its position 2, 3 and/or 4 seeming to be important for the activity. It could highlight compounds 4e and 4f which seem to be slightly better than NPB (whose IC50 on MCF7 cells isn’t reminded). Moreover, compare 4-Cl and 4-CH3 would be interesting too (4p, 4f and this one), to be sure that 4-Cl is worse. And so, for 4i, the 2,3-dichloro could be better..
Response: We have revised the manuscript according to the suggestion of the reviewer. Some of the NPB analogs were observed to be equipotent or slightly more potent when compared to NPB. We have revised the SAR accordingly.
- In the DFT studies part, no reference is cited as well as no explanation of the principle neither the goal is developed. What about the conclusion of these observations? Could you then predict better structures? The part on the hybridization could be shorter, and the conformation of the piperazine more developed.
Response: We have revised the manuscript accordingly according to the suggestions of the reviewer. With regard to the conformation of piperazine, the neighbouring SP3 carbon exerted influence on it. It was also observed in X-ray studies. This suggestion has also been developed in the revised manuscript.
- What about the toxicity of these compounds? And their Lipinski score?
Response: In silico analyses observed that the compounds were not predicted to be hepatotoxic. Lipinski scores are found to be more or less druggable. So we further provided toxicity profile.
- Why did you choose 4s for X ray and crystal packing, the only non-active analogue (and with a 4-Cl)? Be careful, note than 4r in table 1 is the same than 4d (4s doesn’t exist in the experimental section, 4r is mentioned in the conclusion…).
Response: Compound 4s was utilized as an exemplar of the basic structural template of title compounds as the crystals were readily obtained. The typographical error was corrected in the abstract. We have corrected the confusion between 4s/4r/4d. We thank the reviewer for detection of the error.
- The introduction is well documented, but some sentences need to be improved. For a better reading, please mention the structure of NPB, and add its previously reported activity on MCF, as it is the starting point of this new SAR (we can then compare it immediately). For a better comprehension, could you please pick table 1 and store the same colors as in the scheme 1, with the substituents in the same position than compound 4 of scheme 1 (be careful, there is an inversion between R1 and R3 in the legend of scheme 1). Same comment for figures 2 and 3.
Response: We have revised the manuscript according to the reviewer’s suggestion.
- Please, propose a new sentence for l,46, to prevent the repetition of phosphorylation and analogues.
Response: We have revised accordingly.
- L148 check the order, with 4r (the right one?)
Response: This has been corrected. We than k the reviewer for the detection of the error.
- L149 it’s a tolyl, not a benzyl…
Response: We are grateful for the detection of the error and have revised.
- L163 evaluate and confirm? please conclude
Our Response: concluded
- A careful re-reading is necessary (ex l54 “cticially”, l 63 “. .”, l67 “, ,”…some words are cut, numbers not in bold). The experimental part is not enough rigorously described: some 1H are missing (20 H for cpd 4h, 4 H for cpds 4i and 4m, 5 H for cpd 4m without forgetting the form (name cut, spaces, punctuations, bold, only some compounds mention “characterization of” (the name is enough).
Response: We have corrected the above errors.
- Moreover, for the 1H description, I’m surprised by the coupling value J =4Hz. I’m OK for the thiophene, but it in my mind, it can’t be representative of two vicinal protons in a phenyl, if not, could you please justify. In most of your compounds, I don’t really see the para system, usually d, 2H, J~8.0Hz / d, 2H, J~8.0Hz due to the symmetry (note the number after the comma, especially when using a 400 MHz spectrometer to help in the assignment). Could you recheck or justify, please, as a for the t integrating for 2H (I don’t think they are equivalent)? And due to the complexity of the piperazine, I don’t think that its protons are singlets…
Our Response: We retook the proton NMR, which shows a coupling value J =4Hz, We have revised the manuscript accordingly. We are unable to explain piperazine shows a singlet, but the molecules were confirmed by mass spectroscopy and single crystal XRD and hence piperazine is present in the compounds. Furthermore, the crystal confirmed that the compounds exhibit a similar piperazine peak.
- In my opinion, 13C NMR spectra are recorded on a 100 MHz, and all the 13C must be mentioned. Anyway you can precise 2 if 2 carbons have the same chemical shift. Moreover, the abbreviations for the NMR signals’ multiplicity are lacking in the description of material and methods.
Response: We have corrected and revised.
- You should cite the patent : Small molecules, including 2-[(phenyl)(piperazin-1-yl)methyl]phenol derivatives and analogs, as inhibitors of Bcl-2-associated death promoter (bad) phosphorylation By: Lobie, Peter Edward; Pandey, Vijay Kumar; Rangappa, Kanchugarakoppal Subbegowda; Bassappa, Aalundi; Mohan, Chakrabhavi Dhananjaya; Rangappa, Shobith; Srinivasa, Venkatachalaiah World Intellectual Property Organization, WO2018194520 A1 2018-10-25
Response: We have now cited the patent application.
Reviewer 2 Report
Along the lines to what some of them previously published on PNAS in 2018, the authors propose the synthesis and cell viability data of NPB analogues.
The authors in the title of their manuscript anticipate an “optimization of NPB analogs” that in fact does not exist. The IC50 values of the cell viability, but of a single cancer cell line, are of the same order of magnitude as NPB. Among the nineteen synthesized compounds, seven of them have IC50 values greater than 100μm and of these three even greater than 500μm. Furthermore, the values of compounds 4g, 4r and 4s are not reported. It is difficult to understand why the authors chose this last compound for recording their X-ray intensity data.
It follows that, in case the IC50 value of 4s was high, all the information obtained from the crystal packing analysis would be useless or even misleading in structure activity relationships.
Moreover, the authors seem to ignore the variations in lipophilicity in the series of the title compounds. Almost all compounds have logP (computed) values between 5 and 7 comparable to that estimated for NPB. The only compounds with logP values compatible with potentially favorable pharmacokinetics are 4k, 4n, 4o and 4p. These compounds all carry a heteroaromatic ring such as pyrimidine, pyrazole, and furyl but their antiproliferative activity is negligible.
Authors should interpret these experimental data and envisage a more robust rationalization of the results.
Finally, considering that from a synthetic point of view the manuscript does not present any news compared to what was published on PNAS in 2018, while the molecular design is conceived exclusively by close analogy with NPB.
Author Response
Reply to the Reviewers Comments
Reviewer 2 comments
- Along the lines to what some of them previously published on PNAS in 2018, the authors propose the synthesis and cell viability data of NPB analogues. The authors in the title of their manuscript anticipate an “optimization of NPB analogs” that in fact does not exist. The IC50 values of the cell viability, but of a single cancer cell line, are of the same order of magnitude as NPB. Among the nineteen synthesized compounds, seven of them have IC50 values greater than 100μm and of these three even greater than 500μm. Furthermore, the values of compounds 4g, 4r and 4s are not reported. It is difficult to understand why the authors chose this last compound for recording their X-ray intensity data.
Response: We have changed the title of the manuscript in accordance with the reviewer’s comment. Compound 4s was utilized as an exemplar of the basic structural template of title compounds as the crystals were readily obtained.
- It follows that, in case the IC50 value of 4s was high, all the information obtained from the crystal packing analysis would be useless or even misleading in structure activity relationships.
Response: As above and the x-ray data provided information related to intramolecular hydrogen bonding and stability irrespective of the substitutions.
- Moreover, the authors seem to ignore the variations in lipophilicity in the series of the title compounds. Almost all compounds have logP (computed) values between 5 and 7 comparable to that estimated for NPB. The only compounds with logP values compatible with potentially favorable pharmacokinetics are 4k, 4n, 4o and 4p. These compounds all carry a heteroaromatic ring such as pyrimidine, pyrazole, and furyl but their antiproliferative activity is negligible. Authors should interpret these experimental data and envisage a more robust rationalization of the results.
Response: We thank the reviewer for the constructive suggestions. In addition to logP, there is a stability order of the molecules that gave better acclivity. We now provided DFT calculations for this purpose. We have revised the manuscript accordingly.
- Finally, considering that from a synthetic point of view the manuscript does not present any news compared to what was published on PNAS in 2018, while the molecular design is conceived exclusively by close analogy with NPB.
Response: Herein, we have obtained a number of compounds equipotent or slightly more potent than NPB which may satisfy other requirements (such as oral bioavailability) for a preclinical candidate. . We have also obtained the SAR through DFT calculation . In addition, the crystal data provided some basic structural template for these series of molecules.

Round 2
Reviewer 1 Report
Please, see the attached file.

Author Response
Reply to the Reviewers Comments
Reviewer 1
- Point 2: there is no discussion or comparison of 4a, NPB, 4j and 4b, 4g, 4q, which could be interesting to mention, even if it seems too difficult to conclude (additional effects? Cf point 2)
Our response: revised slightly.
- Point 6: For R2 and R3, you could keep the phenyl, which are always present, and just highlight then the substituent changes as below, and apply the same presentation for an easier reading of the table 1 (I don’t see any modification in the table, even for 4r that you told you have taken account...points 5 and 6 of your answer) :
Our response: We revised the manuscript considering the above suggestion.
- Point 4: what about the toxicity on healthy cells?
Our response: The compounds 4e, 4f, 4i, and NPB inhibited the proliferation of normal breast cell MCF10A, with an higher IC50 values of 33.8, 61.4, 28.5, and 110.6 µM, respectively. Incorporated the data after the experiment
- Point 5: I understand that 4s was chosen as a basic structural template of title compounds. But as the crystal structure of 4s shows H bonds with the Cl in para, are you sure that there is the same H bond for the potent cpds 4e and 4f, where the Cl are in positions 2 and 3 (molecular modeling) ? And what about the CH3 of the tolyl? No H bond, while 4i is potent. And I agree for the conformation, as all the substituents R1 and R2 are aromatics (that could be told),and so should adopt the same conformation (even if the heteroatom could involve another one ?).
Our response: After the synthesis of title compounds, we will try to grow crystals for all compounds to get the absolute structural information for the respective series of molecules. Unfortunately, only compound 4s got crystallized and therefore we analyzed. We have tried different solvent combination to get crystals of remaining compounds but we are unable to get crystals of those molecules. therefore, we are failed to represent the crystal of remaining compounds.
- L 172 benzyl has not been replaced by tolyl...Moreover, the structures of 4d and 4r are still the same in table 1 and no 4s appears neither in the experimental section, neither in the Supporting Part.
Our response: Revised Accordingly.
- Point 12, the 1H shows an “m”, not an “s” for the piperazine (large signal)....and 1H are still missing. Moreover, zoom on the aromatic area would help the assignment (I find around 6 Hz for the triplets at 7.17 and 7.52 of 4i)
Our response: We retake the 1H NMR, now we revised manuscript accordingly. We don’t know why piperazine shows singlet, but molecules are confirmed by mass spectroscopy, 13C NMR and also purity of compound is confirmed by HPLC.
- Point 13 hasn’t been modified in my mind.... Ex: L 230, we should see 117.24 (2C), 128.9 (2C)
Our response: Corrected accordingly.
- L 116, careful I think it is a piperazine and not a piperidine
Our response: Revised Accordingly
- Some references were added, but I think that the most ones of 11-14 are the authors ’ones, and are not really necessary.
Our response: reference 11-14 are related to anticancer activity (MCF-7) that’s why we are added.

Reviewer 2 Report
The authors in this revised version present cosmetic changes to the manuscript without substantially responding to the remarks. For these reasons, the overall judgment remains unchanged.
Author Response
Reply to the Reviewers Comments
Reviewer 2 comments
- Along the lines to what some of them previously published on PNAS in 2018, the authors propose the synthesis and cell viability data of NPB analogues.The authors in the title of their manuscript anticipate an “optimization of NPB analogs” that in fact does not exist. The IC50values of the cell viability, but of a single cancer cell line, are of the same order of magnitude as NPB. Among the nineteen synthesized compounds, seven of them have IC50 values greater than 100μm and of these three even greater than 500 μm. Furthermore, the values of compounds 4g, 4r and 4s are not reported. It is difficult to understand why the authors chose this last compound for recording their X-ray intensity data.
Response: We have changed the title of the manuscript in accordance with the reviewer’s comment. In addition, the present manuscript describes the newer version of NPB, where in which we identified the better versions of of diverse structure, whose activity is comparable to NPB. The interdisciplinary work related to in silico DFT calculations of NPB analogs is novel. The single crystal X-ray crystallographic studies of Compound 4r was utilized as an exemplar of the basic structural template of NPB, which gave a new conformational acquirement of piperazine, when compares to its normal structure.
- It follows that, in case the IC50value of 4rwas high, all the information obtained from the crystal packing analysis would be useless or even misleading in structure activity relationships.
Response: Compound 4r x-ray crystal data provided the information related to intra-molecular hydrogen bonding, which gave stability is novel. These observation is important because of the phenol group in vivo generally have tendency to get metabolized.
- Moreover, the authors seem to ignore the variations in lipophilicity in the series of the title compounds. Almost all compounds have logP (computed) values between 5 and 7 comparable to that estimated for NPB. The only compounds with logP values compatible with potentially favorable pharmacokinetics are 4k, 4n, 4o and 4p. These compounds all carry a heteroaromatic ring such as pyrimidine, pyrazole, and furyl but their antiproliferative activity is negligible. Authors should interpret these experimental data and envisage a more robust rationalization of the results.
Response: We thank the reviewer for the constructive suggestions. In addition to logP, there is a stability order of the molecules that gave better acclivity. We now provided DFT calculations for this purpose. We have revised the manuscript accordingly. In addition, the present NPB analogs were found to be selectively cytotoxic to MCF-7 cells when compared to normal MCF-10A cells.
- Finally, considering that from a synthetic point of view the manuscript does not present any news compared to what was published on PNAS in 2018, while the molecular design is conceived exclusively by close analogy with NPB.
Response: Herein, we have obtained a number of compounds equipotent or slightly more potent than NPB which may satisfy other requirements (such as oral bioavailability) for a preclinical candidate. We have also obtained the SAR through DFT calculation. In addition, the crystal data provided some basic structural template for these series of molecules. We wish to write here that we are presently going for IND filing related to this drug discovery program.

Round 3
Reviewer 1 Report
Please, See the file attached below

Author Response
We thank the referee for his valuable suggestions. we revised according to the corrections made by the referee

Reviewer 2 Report
The manuscript has been improved and I believe it can be accepted in its present form
Author Response
We thank the referee for accepting our to publish in IJMS.